

**Coccolithophore biodiversity controls carbonate export in the Southern Ocean**
Andrés S. Rigual Hernández[1,*] Thomas W. Trull[2,3], Scott D. Nodder[4], José A. Flores[1],
Helen Bostock[4,5], Fátima Abrantes[6,7], Ruth S. Eriksen[2,8], Francisco J. Sierro[1], Diana M.
Davies[2,3], Anne-Marie Ballegeer[9], Miguel A. Fuertes[9], and Lisa C. Northcote[4].
1 Área de Paleontología, Departamento de Geología, Universidad de Salamanca, 37008
Salamanca, Spain
2 CSIRO Oceans and Atmosphere Flagship, Hobart, Tasmania 7001, Australia
3 Antarctic Climate and Ecosystems Cooperative Research Centre, University of
Tasmania, Hobart, Tasmania 7001, Australia
4 National Institute of Water and Atmospheric Research, Wellington 6021, New Zealand
5 University of Queensland, Brisbane, Queensland 4072, Australia
6 Portuguese Institute for Sea and Atmosphere (IPMA), Divisão de Geologia Marinha
(DivGM), Rua Alferedo Magalhães Ramalho 6, Lisboa, Portugal
7 CCMAR, Centro de Ciências do Mar, Universidade do Algarve, Campus de Gambelas,
8005-139 Faro, Portugal
8 Institute for Marine and Antarctic Studies, University of Tasmania, Private Bag 129,
Hobart, Tasmania 7001, Australia
9 Departamento de Didáctica de las Matemáticas y de las Ciencias Experimentales,
Universidad de Salamanca, 37008 Salamanca, Spain.
* Corresponding author
**Abstract**
Southern Ocean waters are projected to undergo profound changes in their
physical and chemical properties in the coming decades. Coccolithophore blooms in the
Southern Ocean are thought to account for a major fraction of the global marine calcium
carbonate ($CaCO_3$) production and export to the deep sea. Therefore, changes in the
composition and abundance of Southern Ocean coccolithophore populations are likely to
alter the marine carbon cycle, with feedbacks to the rate of global climate change.
However, the contribution of coccolithophores to $CaCO_3$ export in the Southern Ocean is
uncertain, particularly in the circumpolar Subantarctic Zone that represents about half of
the areal extent of the Southern Ocean and where coccolithophores are most abundant.
Here, we present measurements of annual $CaCO_3$ flux and quantitatively partition them
amongst coccolithophore species and heterotrophic calcifiers at two sites representative





of a large portion of the Subantarctic Zone. We find that coccolithophores account for a
major fraction of the annual $CaCO_3$ export with highest contributions in waters with low
algal biomass accumulations. Notably, our analysis reveals that although *Emiliania*
*huxleyi* is an important vector for $CaCO_3$ export to the deep sea, less abundant but larger
species account for most of the annual coccolithophore $CaCO_3$ flux. This observation
contrasts with satellite remote sensing images that mostly reflect *E. huxleyi* blooms as a
result of its higher cell abundance and detachment of its relatively small liths. It appears
likely that the climate-induced migration of oceanic fronts will initially result in the
poleward expansion of large coccolithophore species increasing $CaCO_3$ production.
However, subantarctic coccolithophore populations will eventually diminish as
acidification overwhelms those changes. Overall, our analysis emphasizes the need for
species-centred studies to improve our ability to project future changes in phytoplankton
communities and their influence on marine biogeochemical cycles.

**1. Introduction**

The emissions of carbon dioxide ($CO_2$) into the atmosphere by anthropogenic

industrial activities over the past 200 years are inducing a wide range of alterations in the
marine environment (Pachauri et al., 2014). These include ocean warming, shallowing
of mixed layer depths, changes in nutrient supply to the photic zone, and decreasing
carbonate-ion concentrations and pH of the surface ocean, a process known as ocean
acidification (Rost and Riebesell, 2004; Stocker et al., 2014). In particular, ocean
acidification poses a major global-scale risk for marine calcifying organisms because the
decline in the saturation state of carbonate minerals in seawater makes the biological
precipitation of carbonate difficult and increases the dissolution rates of their shells or
skeletons (Gattuso and Hansson, 2011). Owing to their moderate alkalinity and cold
temperatures, Southern Ocean waters are projected to become undersaturated with respect
to aragonite no later than 2040 and to calcite by the end of the century (Cao and Caldeira,
2008; McNeil and Matear, 2008; Shadwick et al., 2013). Since such thresholds will be
reached sooner in polar regions, Southern Ocean ecosystems have been proposed as
bellwethers for prospective impacts of ocean acidification on marine organisms at mid
and low latitudes (Fabry et al., 2009).

Coccolithophores are a major component of phytoplankton communities in the

Southern Ocean, particularly in its northern-most province, the Subantarctic Zone, where





they often exhibit maximum abundances and diversity (e.g. Gravalosa et al., 2008;
Saavedra-Pellitero et al., 2014; Malinverno et al., 2015; Charalampopoulou et al., 2016).
Coccolithophores play an important and complex role in the Southern Ocean carbon cycle
(Salter et al., 2014). On the one hand, the production of calcite platelets (termed
coccoliths) decreases the alkalinity of surface waters thereby reducing the atmospheric
uptake of $CO_2$ from the atmosphere into the surface ocean. On the other hand, the
production of organic matter through photosynthesis, and its subsequent transport to
depth in settling particles, enhances carbon sequestration via the biological carbon pump
(Volk and Hoffert, 1985). Additionally, due to their high density and slow dissolution,
coccoliths act as an effective ballast for organic matter, increasing organic carbon
sequestration depths (Buitenhuis et al., 2001; Boyd and Trull, 2007; Ziveri et al., 2007).
Therefore, changes in the abundance, composition and distribution of coccolithophores
could have an extensive impact on ocean nutrient stoichiometry, carbon sequestration,
and nutrition for higher trophic levels in the Southern Ocean (Deppeler and Davidson,

2017).

The remoteness and vastness of the Southern Ocean, together with the inherent

temporal and spatial variability of pelagic ecosystems, hampers accurate characterization
and quantification of Southern Ocean phytoplankton communities. Advances in satellite
technology and modelling algorithms have allowed a circumpolar and year-round
coverage of the seasonal evolution of major phytoplankton functional groups within the
Southern Ocean (e.g. Alvain et al., 2013; Hopkins et al., 2015; Rousseaux and Gregg,
2015). In particular, satellite reflectance observations have been used to quantitatively
estimate coccolithophore Particulate Inorganic Carbon (PIC) concentrations throughout
the Southern Ocean. These satellite estimates suggest apparent high PIC values during
summer near the major Southern Ocean fronts attributed to coccolithophores (Balch et
al., 2011; Balch et al., 2016). This band of elevated reflectance and PIC that encircles the
entire Southern Ocean was termed the "Great Calcite Belt" by these authors. However,
recent research (Trull et al., 2018) indicates that satellite ocean-colour-based PIC
estimates could be unreliable, particularly in Antarctic waters where they erroneously
suggests high PIC abundances. Shipboard observations, on the other hand, provide a
detailed picture of phytoplankton community composition and structure, but are
dispersed, both temporally and geographically, and provide rather heterogenous data in
terms of taxonomic groups investigated, and the sampling scales and methodologies used



(e.g. Kopczynska et al., 2001; de Salas et al., 2011; Poulton et al., 2013; Patil et al., 2017,
among others). *In situ* year-round monitoring of key strategic regions is  critically needed
to establish baselines of phytoplankton community composition and abundance and to
validate and improve ocean biogeochemical models (Rintoul et al., 2012). This
information is also essential if we are to detect possible climate-driven changes in the
structure of phytoplankton communities that could influence the efficiency of the
biological carbon pump, with consequent feedbacks to the rate of deep-water carbon
sequestration and global climate change (Le Quéré et al., 2007; Deppeler and Davidson,

2017).

Here, we document coccolithophore and carbonate particle fluxes collected over

a year by four sediment trap records deployed at two strategic locations of the Australia
and New Zealand sectors of the Southern Ocean considered representative of a large
portion of the SAZ. Our measurements provide coccolith mass estimates of the main
coccolithophore species and quantitatively partition annual carbonate fluxes amongst
coccolithophore species and heterotrophic calcifiers. We find that coccolithophores are a
major vector for $CaCO_3$ export out of the mixed layer and that the largest contribution to
$CaCO_3$ export is not from the most abundant species *Emiliania huxleyi* but rather from
larger coccolithophores species with substantially different physiological traits (e.g.
*Calcidiscus leptoporus*). Our results emphasize the urgent need for diagnostic fitness
response experiments on other coccolithophore species aside from *E. huxleyi* (e.g. Feng
et al., 2017) in order to be able to be able to predict the impacts of anthropogenically
induced changes in Southern Ocean ecosystems and biological carbon uptake
mechanisms.

**2. Material and methods**

**2.1 Oceanographic setting**

The SAZ alone accounts for more than half of the Southern Ocean area (Orsi et

al., 1995) and represents a transitional boundary between the warm, oligotrophic waters
of the subtropical gyres to the north and the cold, silicate-rich waters south of the Polar
Front (PF). The SAZ is arguably the largest high nutrient, low chlorophyll (HNLC)
province in the world's ocean and is central to the linkages between the ocean–
atmosphere $CO_2$ exchange and climate. The deep winter convection in the SAZ, which



that exceeds 400 m, results in the formation of a high-oxygen water masses known as
Subantarctic Mode and Antarctic Intermediate Waters that connect the upper and lower
limbs of the global overturning circulation (Sloyan and Rintoul, 2001a, b). The formation
of these water masses are responsible for the sequestration of a large fraction of
anthropogenic $CO_2$ (Sabine et al., 2004), with an estimated 1 Gt C yr$^{-1}$ transported to
intermediate depths annually (Metzl et al., 1999). Macronutrient concentrations display
pronounced seasonal changes in the SAZ with fully replete levels during winter to
substantial depletion during summer, particularly for silicate (Dugdale et al., 1995;
Rintoul and Trull, 2001; Bowie et al., 2011). Phytoplankton community in the
subantarctic zone is dominated by pico- and nanoplankton including cyanobacteria,
coccolithophores and autotrophic flagellates with lower abundances of diatoms than polar
waters south the Polar Front (Chang and Gall, 1998; Kopczynska et al., 2001; de Salas et
al., 2011; Rigual-Hernández et al., 2015b; Eriksen et al., 2018).

**2.2 Field experiments**
Here we report on the coccolithophore and biogeochemical fluxes collected over
a year at the Australian Southern Ocean Time Series (SOTS) observatory (Trull et al.,
2010) and the New Zealand Subantarctic Mooring (SAM) site (Nodder et al., 2016) (Fig.
1). The SOTS observatory is located in the abyssal plane of the central SAZ
approximately 530 km southwest of Tasmania (46° 56' S, 142° 15' E) within an anti-
cyclonic gyre in a region characterized by weak circulation (Trull et al., 2001; Herraiz-
Borreguero and Rintoul, 2011). SOTS was equipped with three vertically moored, conical
time-series sediment traps (McLane Parflux Mk 7G-21) placed at ~1000, 2000 and 3800
m depth between August 2011 until July 2012. The physical, chemical and biological
parameters of SOTS site are regarded as representative for large portion of the Indian and
Australian sectors of the SAZ (~90°E and 140°E; Trull et al., 2001). The SAM site is
located in the Bounty Trough in in the subantarctic waters south east of New Zealand
(46°40'S, 178' 30°E) and was equipped with conical, time-incremental sediment trap
(McLane PARFLUX Mk7G-21) placed at 1500 m depth, with samples used in the present
study collected between November 2009 until November 2010. The SAM site is
considered to be representative of a wide area of the northern sector of the SAZ off eastern
New Zealand, approximately 171°E to 179°W and 45 to 47°S  (Law et al., 2014; Fig. 1).
Full details of the field experiments from these two localities in the Australian and New



Zealand sectors of the SAZ can be found in Trull et al. (2001) and Nodder et al. (2016),
respectively.

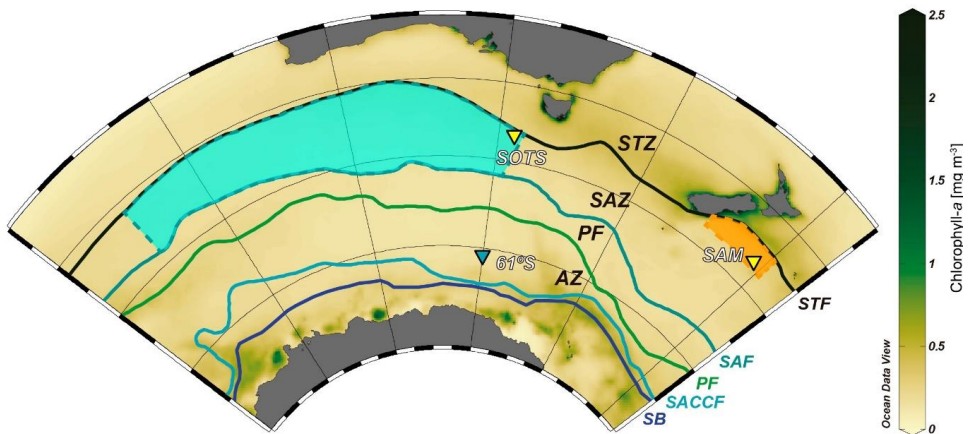


**Figure 1**: Chlorophyll-*a* composite map of the Australian-New Zealand sector of the
Southern Ocean (July 2002 to July 2012) from the MODIS Aqua Sensor showing the
location of the sediment trap moorings sites: SOTS, 61°S and SAM. The regions for
which the SOTS and SAM sites are representative are marked with light blue and orange
areas, respectively. Abbreviations: Subtropical front - STF, Subantarctic Zone – SAZ,
Subantarctic Front - SAF, Polar Frontal Zone - PFZ, Polar Front - PF, Antarctic Zone –
AZ, Southern Antarctic Circumpolar Current Front – SACCF, southern boundary of the
ACC – SB. Oceanic fronts after Orsi et al. (1995).
**2.3 Sample processing**
In short, the recovered trap bottles were refrigerated upon recovery and then
allowed to settle. The sample slurry was then wet-sieved through a 1 mm screen in the
case of SOTS (no attempt to extract zooplankton "swimmers" was made for the <1 mm
fraction analysed here) and through a 200 µm sieve to remove "swimmers" for the SAM
site. The remaining fraction was then split using a McLane wet sample divider; the SOTS
samples were subdivided into one tenth aliquots while one fifth splits were made for the
SAM samples. For the SOTS samples, a total of 55 samples were processed for calcareous
nannoplankton analysis. The one-tenth splits dedicated to phytoplankton analysis were
further subdivided into four aliquots with the McLane splitter. One aliquot was used for





calcareous nannoplankton analysis and the remaining three were kept refrigerated for
biomarker and non-calcareous microplankton analyses. In the case of the SAM samples,
the one-fifth aliquots were further subdivided into five subsplits, and one of those was
used for calcareous nannoplankton analysis. Two different types of glass slides per
sample were prepared. The first preparation was used for the estimation of coccosphere
and calcareous dinocyst (calcispheres of thoracosphaerids) fluxes and for coccolith
imaging. A volume ranging between 1000 and 5000 µl of the raw sample was mounted
on a glass slide using Canada balsam following Flores and Sierro (1997). This technique
produces random settling of the coccoliths for microscopic identification and
enumeration. The second type of glass slide was prepared following a modified protocol
for non-destructive disintegration of aggregates modified from Bairbakhish et al. (1999).
The objective of this chemical treatment is to reduce biases in the coccolith flux
estimations associated with the presence of different types of aggregates and
coccospheres (Bairbakhish et al., 1999). In brief, 2000 µl were extracted from the aliquot
for calcareous nannoplankton analysis and then treated with a solution comprising 900 µl
sodium carbonate and sodium hydrogen carbonate, 100 µl ammonia (25%) and 2000 µl
hydrogen peroxide (25%). The sample was agitated for 10 seconds every 10 minutes and
this process was repeated over an hour. Then, the reaction was stopped with catalase
enzyme and samples were allowed to settle for at least 48 hours before preparation on
microscope slides. pH controls indicate that the solution kept pH levels near 9, therefore
precluding coccolith dissolution. Finally, trap samples were mounted on microscope
slides following the same decantation method as used for the first type of glass slides (i.e.
Flores and Sierro, 1997).
**2.4 Determination of CaCO₃ fluxes**

A detailed description of the geochemical analytical procedures for the SOTS

samples is provided in Trull et al. (2001) and Rigual-Hernández et al. (2015a) while more
detailed procedures of the SAM trap can be found in Nodder et al. (2016). In short, for
the SOTS site three of the one tenth splits were filtered onto 0.45 pore size filters. Then
the material was removed from the filter as a wet cake of material, dried at 60°C, and
ground in an agate mortar. This material was used to determine the total mass and
composition of the major components of the flux. Particulate inorganic Carbon (PIC)
content was measured by closed system acidification with phosphoric acid and
coulometry. For the SAM site, one-fifth split was analysed for elemental calcium (Ca)





concentration using ICP-MS techniques. The samples were oven-dried, digested in
nitric/hydrochloric acid and then analysed according to the methods under US EPA 200.2.
Ca was used to estimate $CaCO_3$ content in the samples assuming a 1:1 molar ratio in
$CaCO_3$.


**2.5 Quantification and characterization of coccolithophore sinking assemblages**

Qualitative and quantitative analyses of coccospheres and coccoliths were

performed using a Nikon Eclipse 80i polarised light microscope at 1000 x magnification.
The taxonomic concepts of Young et al. (2003) and the Nannotax website (Young et al.,
2019) were used. A target of 100 coccospheres and 300 coccoliths was established;
however, owing to the pronounced seasonality in coccolithophore export, there were
some periods with very low abundance of coccospheres in the samples and therefore the
target of 100 coccospheres was not always met. Coccosphere and coccolith species counts
were then transformed into daily fluxes after Rigual Hernández et al. (2018).

**2.6 Determination of coccolith mass and size**

Birefringence and morphometric methods are the two most commonly used

approaches for estimating the calcite content of isolated coccoliths. The circularly-
polarized light-microscopy-based technique (Fuertes et al., 2014) is based on the
systematic relationship between the thickness of a given calcite particle (in the thickness
range of 0 - 1.55 mm) and the first-order polarization colours that it displays under
polarized light (Beaufort, 2005; Beaufort et al., 2014; Bolton et al., 2016). The advantages
of this approach are that: (i) it directly measures complete coccoliths with no assumptions
regarding their shape or thickness and (ii) it allows for quantification of calcite losses
associated with missing parts or etching of the coccoliths. Disadvantages of this technique
are the errors associated with the coccolith-calcite calibration and their consequent effect
on the coccolith mass estimates (Fuertes et al., 2014; González Lemos et al., 2018). The
morphometric approach, on the other hand, allows better taxonomic identification of the
coccoliths and has smaller errors in the length measurements (~0.1 to 0.2 μm; Poulton et
al. 2011). However, this method does not allow direct measurement of coccolith thickness
and assumes identical shape and width proportions for all specimens of the same species,
among other uncertainties (see Young and Ziveri, 2000 for a review). Since the two



methods have different associated errors (Poulton et al., 2011), we applied both
approaches to our coccolith flux data in order to obtain two independent estimates of the
fractional contribution of coccolithophores species to total carbonate export in the SAZ.
For the birefringence-based approach, a minimum of 50 coccoliths of each of the
main coccolithophore species were imaged using a Nikon Eclipse LV100 POL light
microscope equipped with circular polarisation and a digital camera (Nikon DS-Fi1 8-bit
colour). The only exception was *E. huxleyi* for which coccolith mass values had already
been estimated in all the same samples at high resolution by Rigual-Hernández et al.
(under review). For the minor components of the flux assemblage, a lower number of
coccoliths were measured (Table 1). A photograph of the same apical rhabdolith of the
genus *Acanthoica* was taken and used for calibration at the beginning of each imagining
session during which microscopy light and camera settings were kept constant. A
different number of fields of view of multiple samples representative of different seasons
were photographed until the target number of coccoliths for each species was reached.
Photographs were then analysed by the image processing software C-Calcita. The output
files for single coccoliths were visually selected and classified into the lowest possible
taxonomic level. Length and weight measurements were automatically determined by C-
Calcita software. Morphometric measurements of all the species are summarized in Table
1. For further methodological details see Fuertes et al. (2014) and Bolton et al. (2016).
The second approach consisted of performing morphometric measurements on the
coccoliths followed by the estimation of their coccolith mass assuming a systematic
relation between length and thickness (Young and Ziveri, 2000). Young and Ziveri (2000)
proposed that the calcite content of a given coccolith could be estimated using the
following formula:
$$\text{Coccolith calcite (pg)} = 2.7 \times k_s \times l^3$$
where 2.7 is the density of calcite ($CaCO_3$; pg $\mu m^3$), "$k_s$" is a shape constant that varies
between species and morphotypes and whose value is based on the reconstruction of
coccolith cross profiles and "l" is the distal shield length (DSL). In order to undertake
coccolith measurements on the same coccoliths used for the birefringence-based
approach, we employed the distal shield length values measured by C-Calcita using
circularly polarized light instead of morphometric measurements on Scanning Electron
Micrographs (SEM) as made in Young and Ziveri (2000).

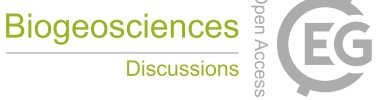

Since coccolith distal shield length (DSL) has been reported to be systematically
underestimated using cross-polarized light microscopy (e.g. D'Amario et al., 2018), we
evaluated the possible errors in the DSL measurements made by C-Calcita. For this
assessment, we measured 40 detached coccoliths of *C. leptoporus* under the SEM from
samples of the SOTS sediment traps using the image processing software Image-J.
Average DSL measurements under the SEM were then compared with those made by C-
Calcita on 40 randomly selected *C. leptoporus* coccoliths. The average coccolith length
obtained with the SEM analysis ($6.37 \pm 1.02$, n = 40) was ~ 4% shorter than that estimated
with C-Calcita ($6.62 \pm 1.47$, n = 40). Therefore, we assumed the error for the DSL
measurements with circularly polarized light is < 5%. For the $k_s$ value of each taxa, data
from the literature were (Table 1). *E. huxleyi* assemblages in the SAZ are composed of a
mixture of five different morphotypes: A, A overcalcified, B, B/C and C, each of which
is characterized by different shape factors ($k_s$). Since $k_s$ is not available for all the
morphotypes found in the SAZ and it is not possible to differentiate between morphotypes
in our light microscopy images, we used the mean shape factor constant for *E. huxleyi*
(i.e. $k_s = 0.0275$) to provide a range of coccolith mass estimates for this species (Table 1
and Fig. 4).
**2.7 Calculation of annual estimates**
Since the trap collection periods encompassed a period shorter than a calendar
year, annual estimates of coccolith and $CaCO_3$ fluxes had to be estimated. For the SOTS
site, a total of 336 days were sampled for the 1000 and 2000 m traps and 338 days for the
3800 m. Since the unobserved interval occurred in winter, the missing sampling period
was filled using an average flux value of the winter cups (first and last trap bottles). In
the case of the SAM trap, the number of samples available for $CaCO_3$ and calcareous
nannoplankton analyses was different, covering a period of 313 and 191 days
respectively. Since gaps were quasi-equally distributed along the time series, annual
fluxes were estimated by filling the gaps in the record with average fluxes calculated from
the available data. The estimated range of the annual contribution of coccolithophores to
total $CaCO_3$ export at the SOTS and SAM traps was calculated by multiplying the
coccolith flux of each species in each sampling interval by its average coccolith weight
values obtained with the birefringence and morphometric techniques.
**3. RESULTS**



### 3.1 Magnitude and seasonality of coccolithophore fluxes

Annualized coccolith fluxes were similar at the SOTS three trap depths, with 8.6, 7.3 and 8.6 x $10^{11}$ liths m$^{-2}$ yr$^{-1}$ at 1000, 2000 and 3800 m respectively, and about three times larger than those of the SAM site (3.0 x $10^{11}$ liths m$^{-2}$ yr$^{-1}$). The contribution of intact coccospheres to the total coccolith export was low at both sites, with annual coccosphere fluxes two orders of magnitude lower than coccolith fluxes at SOTS (3.5, 3.3 and 1.8 x $10^9$ coccospheres m$^{-2}$ yr$^{-1}$ at 1000, 2000 and 3800 m, respectively) and SAM (2.2 x$10^9$ coccospheres m$^{-2}$ yr$^{-1}$).

Both coccolith and coccosphere fluxes displayed a marked seasonality that followed the general trend of algal biomass accumulation in the surface waters at the SOTS and SAM sites (Fig. 2). Coccolith fluxes at 1000 m started to increase in early October and remained above the threshold of 1 x $10^9$ coccoliths m$^2$ d$^{-1}$ until mid-April, i.e. approximately eight months (Fig. 2). Three maxima were recorded during the period of high coccolith export: October-early November 2011 (4 x $10^9$ coccoliths m$^2$ d$^{-1}$), late December 2011 (9 x $10^9$ coccoliths m$^2$ d$^{-1}$) and March 2012 (4 x $10^9$ coccoliths m$^2$ d$^{-1}$). Coccolith fluxes of the main coccolithophore species generally followed the similar seasonal pattern to that of the total coccolith flux (Supplementary figure 1) and are not discussed further. Coccolithophore fluxes registered by the 2000 and 3800 m sediment traps followed a generally similar seasonal pattern to those of the shallower trap at the SOTS site (Fig. 2). At SAM, coccolith fluxes exhibited a strong seasonality with peak fluxes in early January 2010 (up to 6 x $10^9$ coccoliths m$^2$ d$^{-1}$) and a secondary peak in August 2010 (3 x $10^9$ coccoliths m$^2$ d$^{-1}$). Coccosphere fluxes at both sites displayed maximum fluxes during the austral summer and minima during winter; however maximum coccosphere export peaks did not always match those of coccolith export (Fig. 2).

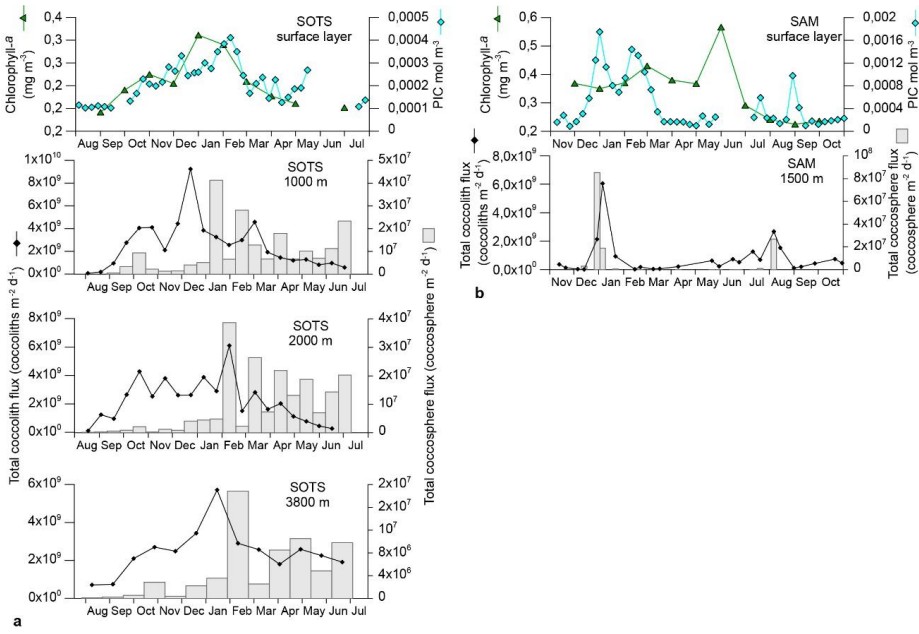

**Figure 2:** Satellite-derived chlorophyll-*a* and Particulate Inorganic Carbon (PIC) concentration in the surface layer and total coccolith and coccosphere fluxes registered by the sediment traps at the SOTS (a), SAM (b) and 61°S sites (c, Rigual Hernández et al., 2018). Coccosphere fluxes are not available for the 61°S site.

### 3.2. Coccolithophore assemblage composition

Coccolith sinking assemblages were overwhelmingly dominated by *Emiliania huxleyi* for all sediment trap records analysed (Fig. 3a). At the SOTS site, the annualized integrated relative contribution of *E. huxleyi* decreased slightly with depth, comprising 88% of the total coccolithophore assemblage at 1000 m, 82% at 2000 m and 80% at 3800 m. Secondary components of the coccolith sinking assemblage were *Calcidiscus leptoporus* (*sensu lato*) (6.7, 10.2 and 9.7% at 1000, 2000 and 3900 m, respectively), *Heliscosphaera carteri* (1.4, 2 and 1.4%) and small *Gephyrocapsa* spp. (< 3 µm) (1.4, 1.5 and 4.4%). Background concentrations (≤ 1%) of *Calciosolenia* spp.*, Coccolithus pelagicus*, *Gephyrocapsa muellerae*, *Gephyrocapsa oceanica*, *Gephyrocapsa* spp. (> 3 µm), *Syracosphaera pulchra*, *Syracosphaera* spp., *Umbellosphaera tenuis* (*sensu lato*), and *Umbilicosphaera sibogae* were also registered. At the SAM site, *E. huxleyi* accounted for 83% of the annualized coccolith flux, with subordinate contributions of *C. leptoporus* (12.2%) and *Gephyrocapsa* spp. (< 3 µm) (1.5%). Background concentrations (< 1%) of



*Calciosolenia* spp., *G. oceanica*, *Gephyrocapsa* spp. (> 3 µm), *H. carteri*, *Syracosphaera*
spp., *U. sibogae* and *U. tenuis* were observed.

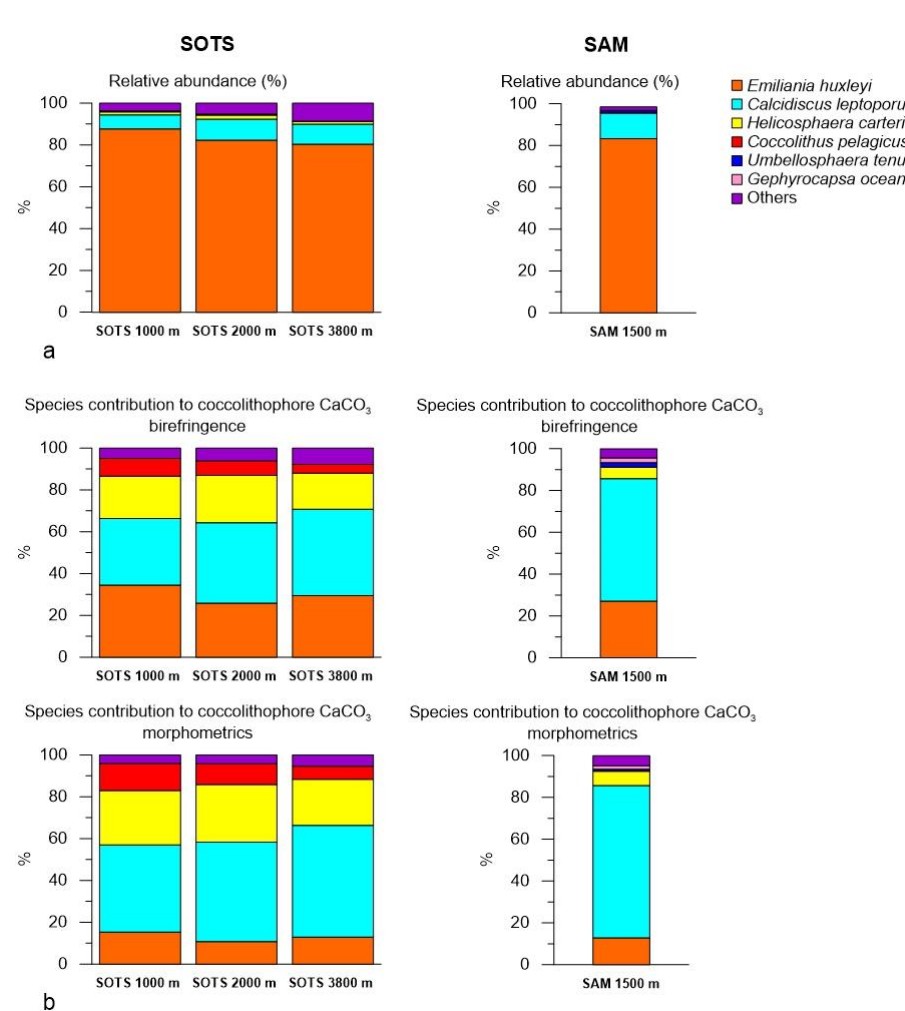

a

b

**Figure 3: a**. Annualized integrated relative abundance of the most important
coccolithophore species in the SOTS and SAM sediment trap records. **b**. Fractional
contribution of coccolithophore species to total coccolithophore CaCO$_3$ in the SOTS and
SAM sediment traps.
**3.3 Calcite content per species**
Coccolith length and mass for all species measured using birefringence and
morphometric techniques are provided in Table 1. Overall, the average coccolith mass
estimates for the coccolithophore species at SOTS and SAM sites using both approaches



are within the range of values in the published literature. The Noelaerhabdaceae family
members, *G. oceanica* and *Gephyrocapsa* spp., display almost identical mass values with
both approaches (Fig. 4). In contrast, substantial discrepancies are identifiable for *C.*
*pelagicus*, *C. leptoporus*, *H. carteri* and *U. sibogae,* for which coccolith mass estimates
are about two-fold greater using morphometrics than with the birefringence approach.
The range of annual contributions of coccolithophores to carbonate is illustrated in Figure

5.

| Species and morphotypes | Type of measurement | n | Length (µm) | | Mass CaCO₃ (pg) | | $k_s$ | Crystal units types | References |
|---|---|---|---|---|---|---|---|---|---|
| | | | Average | SD | Average | SD | | | |
| *Calcidiscus leptoporus* | Birefrigence | 210 | 6.39 | 1.49 | 33.65 | 21.11 | - | V and R | |
| | Morphometrics | 210 | 6.39 | 1.49 | 66.23 | 53.28 | 0.080 | | 1 |
| | Literature estimates | - | 4.3-9.6 | | 22.6-125.2 | | 0.061-0.105 | | 1,2 |
| *Coccolithus pelagicus* | Birefrigence | 54 | 13.28 | 1.14 | 170.90 | 32.33 | - | V and R | |
| | Morphometrics | 54 | 13.28 | 1.14 | 387.96 | 99.64 | 0.060 | | 1 |
| | Literature estimates | - | 8.5-13.5 | | 99.5-398.6 | | 0.051 - 0.060 | | 1,2,3 |
| *Emiliania huxleyi* | Birefrigence | 12842 | 2.78 | 0.57 | 2.64 | 1.43 | - | R | |
| | Morphometrics | 12842 | 2.78 | 0.57 | 0.99-2.64 (1.81)* | 0.60-1.60 | 0.015-0.04 (0.0275)* | (V-unit vestigial) | |
| *E. huxleyi* type A | Literature estimates | - | 3-4 | | 1.50 - 3.50 | | 0.02 | | 1,4,5 |
| *E. huxleyi* type A o/c | Literature estimates | - | 3.5 | | 4.6 | | 0.04 | | 1 |
| *E. huxleyi* type B/C | Literature estimates | - | 1.8-5.5 | | 0.3-3.5 | | 0.015 | | 5,6,7 |
| *E. huxleyi* type B | Literature estimates | - | 3.5-5 | | 2.30 - 6.81 | | 0.02 | | 1,5 |
| *Gephyrocapsa oceanica* | Birefrigence | 51 | 5.87 | 0.62 | 26.70 | 5.64 | - | R | |
| | Morphometrics | 51 | 5.87 | 0.62 | 28.14 | 8.97 | 0.050 | (V-unit vestigial) | |
| | Literature estimates | - | 5-5.35 | | 16.9-25.7 | | 0.050-0.062 | | 1,2 |
| *Gephyrocapsa* spp. | Birefrigence | 10 | 4.03 | 0.59 | 9.00 | 2.51 | - | R | |
| | Morphometrics | 10 | 4.03 | 0.59 | 9.33 | 3.84 | 0.050 | (V-unit vestigial) | 1 |
| | Literature estimates | - | - | - | - | | - | | |
| *Helicosphaera carteri* | Birefrigence | 64 | 11.20 | 1.12 | 100.10 | 20.34 | - | V and R | |
| | Morphometrics | 64 | 11.20 | 1.12 | 194.95 | 56.45 | 0.050 | | 1 |
| | Literature estimates | - | 9.1-10 | | 135-142.8 | | 0.050-0.070 | | 1,2 |
| *Syracosphaera pulchra* | Birefrigence | 81 | 6.77 | 1.09 | 17.77 | 6.80 | - | V, R and T | |
| | Morphometrics | 81 | 6.77 | 1.09 | 26.94 | 11.16 | 0.030 | | 1 |
| | Literature estimates | - | 2.7-6 | | 13.5-16.5 | | 0.027-0.083 | | 1,2,4 |
| *Umbellosphaera tenuis* | Birefrigence | 54 | 6.42 | 0.99 | 15.69 | 5.02 | - | R | |
| | Morphometrics | 54 | 6.42 | 0.99 | 11.45 | 4.61 | 0.015 | | 1 |
| | Literature estimates | - | 5-6 | | 8.7-23.9 | | 0.015-0.071 | | 1,2 |
| *Umbilicosphaera sibogae* | Birefrigence | 6 | 7.76 | 1.81 | 27.14 | 11.07 | - | V and R | |
| | Morphometrics | 6 | 7.76 | 1.81 | 78.93 | 51.38 | 0.055 | | 1 |
| | Literature estimates | - | 4.1-6 | - | 16-35 | | 0.055-0.086 | | 1,2 |

**Table 1**: Coccolith mass estimates of the main coccolithophore species found at the SOTS
and SAM sites using birefringence (C-Calcita) and morphometrics. Additionally, length
and mass estimates from the literature are also listed for most species. References: (1)
Young and Ziveri (2000), (2) Beaufort and Heussner (1999), (3) Samtleben and Bickert
(1990), (4) Poulton et al. (2010), (5) Poulton et al. (2011), (6) Holligan et al. (2010) and
(7) Charalampopoulou et al. (2016). * coccolith mass range obtained applying the
minimum and maximum $k_s$ values for *E. huxleyi* found in the literature (i.e. 0.015 and
0.04, respectively).



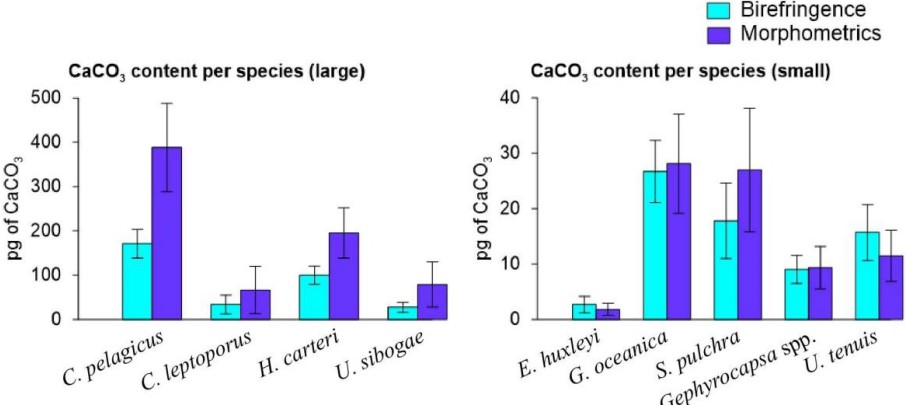


**Figure 4**: Average and standard deviation of the coccolith mass estimates of the most important coccolithophore species captured by the SOTS and SAM sediment traps using birefringence (C-Calcita) and morphometric approaches. For *E. huxleyi,* the morphometric-based coccolith mass estimate was calculated by applying a mean shape factor constant ($k_s$) value estimated from the range of all the morphotypes found at the SAZ (i.e. $k_s = 0.0275$, Table 1).

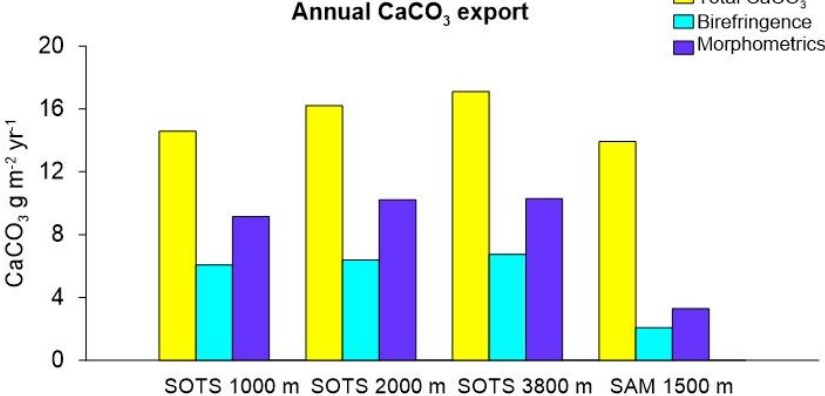

**Figure 5:** Inorganic carbon ($CaCO_3$) fluxes of coccolith calcite estimated using birefringence (C-Calcita) and morphometric approaches for the SOTS and SAM sites.

**4. Discussion**

**4.1 Coccolithophore phenology in the SAZ: satellite versus sediment trap records**

Total coccolith flux seasonality at the SOTS site shows good congruence with satellite-derived PIC in the surface layer, with both parameters suggesting enhanced





coccolithophore productivity between October and March (austral mid-spring to early
autumn; Fig. 2a). Interestingly, substantial coccosphere export ($> 1 \times 10^7$ coccospheres
$m^2 d^{-1}$) does not occur until January indicating that coccolith and coccosphere export are
not tightly coupled in the subantarctic waters south of Australia. Two different processes
could be invoked to explain the mismatch between coccolith and coccosphere fluxes at
this site. Firstly, *E. huxleyi*, the dominant coccolithophore species in the Southern Ocean,
is able to produce coccoliths rapidly (up to three coccoliths per hour; Paasche, 1962;
Balch et al., 1996) and shed the excess of coccoliths into the surrounding water under
certain environmental conditions (Paasche, 2002). Although the coccolith shedding rate
of *E. huxleyi* increases linearly with cellular growth rate (Fritz and Balch, 1996; Fritz,
1999), the tiny size and low weight of detached coccoliths allow them to remain in the
upper water column long after cell numbers have begun to decline. It follows that high
concentrations of detached coccoliths do not necessary imply a proportional abundance
of coccospheres in the surface layer (Tyrrell and Merico, 2004; Poulton et al., 2013) or in
the traps. Additionally, a substantial fraction of the coccospheres produced in the surface
layer may experience substantial mechanical breakage by zooplankton before reaching
the trap depths. Indeed, previous studies in the subantarctic waters south of Tasmania
demonstrated that microzooplankton grazing pressure is sufficient to remove up to 82%
of primary production in mid-summer (Pearce et al., 2011) and most of the particles
exported out the mixed layer during the productive period occur in the form of faecal
aggregates (Ebersbach et al., 2011). Therefore, it is highly likely that: (i) the intensity of
coccosphere export registered by the traps is influenced by grazing pressure in the surface
layer, and (ii) that the impact of grazing on coccolithophores varies throughout the year
(Calbet et al., 2008; Lawerence and Menden-Deuer, 2012; Quéguiner, 2013).
In contrast, seasonal variations in satellite-derived PIC concentration and
coccolith fluxes at SAM show some discrepancies not observed at SOTS. While
maximum PIC concentrations in the surface layer and coccolith and coccosphere fluxes
co-occur in December and January (austral early to mid-summer), satellite-derived PIC
suggests a secondary maximum in February-early-March not recorded by the trap (Fig.
2b). One possibility is that the satellite secondary maximum is not coccoliths. The higher
chlorophyll-a levels at the SAM site (Fig. 2) suggests that other phytoplankton groups,
such as diatoms, are more abundant than in the subantarctic waters south of Tasmania.
Empty and broken diatom valves have been suggested to display similar spectral
characteristics than those of coccolithophore blooms (Broerse et al., 2003; Tyrrell and

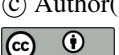



Merico, 2004; Winter et al., 2014). Therefore, the second peak in satellite-derived PIC
could have been caused by a senescent diatom bloom. This hypothesis is likely since
diatom blooms in the SAZ are known to develop early in the productive season (Rigual-
Hernández et al., 2015b) and rapidly decay following the depletion of silicate and/or iron
stocks in the surface layer (Lannuzel et al., 2011). However, no secondary late summer
maximum was observed in biogenic silica fluxes in the SAM. Another possible
explanation is a contribution to the satellite record from lithogenic material.  Fully
resolving causes of mismatches between *in-situ* and satellite PIC estimates is not
achievable for the SAM site (nor more broadly for the Southern Ocean; Trull et al., 2018).
A second difference between the SAM and SOTS sites is that maximum annual
coccosphere export occurred one week earlier than maximum coccolith fluxes at SAM,
(Fig. 2). The different seasonalities between the sites suggest that different export
mechanisms may operate. The formation of rapidly sinking algal aggregates by diatoms
is known to scavenge particles they have collided with and increase particle sinking
(Alldredge and McGillivary, 1991; Passow and De La Rocha, 2006), thus the formation
of such rapid-sinking aggregates could potentially facilitate the preservation of
coccospheres early in the productive season at the SAM site. However, the lack of
accompanying *in situ* information on plankton community structure in the study region
precludes the assessment of these hypotheses.
Despite the uncertainties involved in our interpretations, the overall picture that
emerges from our comparison of satellite and sediment trap flux data is that the duration
of the coccolithophore bloom based on ocean-colour-based PIC concentrations most
likely provides an over-estimation of the coccolithophore productive season. Our
observations motivate caution in describing coccolithophore phenology solely based on
satellite-derived PIC concentrations (e.g. Hopkins et al., 2015).

**4.2 Magnitude and composition of subantarctic coccolithophore assemblages**
Annual coccolith export across the major zonal systems of the Australian sector
of the Southern Ocean exhibits a clear latitudinal gradient, with maximum fluxes at the
SAZ (8.6 x $10^{11}$ liths m$^{-2}$ yr$^{-1}$) and eight-fold lower fluxes in the polar waters of the AZ
(1.0 x 1011 liths m-2 yr-1; Rigual-Hernández et al., 2018). Coccolithophore species
occurrence documented by our subantarctic sediments traps are consistent with previous
reports on coccolithophore assemblage compositions in the surface layer (Findlay and



Giraudeau, 2000; Saavedra-Pellitero et al., 2014; Malinverno et al., 2015; Chang and
Northcote, 2016) and sediments  (Findlay and Giraudeau, 2000; Saavedra-Pellitero and
Baumann, 2015) and are more diverse than those found in the AZ (Rigual Hernández et
al., 2018). The southward decline in coccolithophore abundance and diversity is most
likely due to the decrease in sea-surface temperature (SST) and light availability moving
poleward (Charalampopoulou et al., 2016; Trull et al., 2018). In particular, the close
relationship between temperature and growth rates has been demonstrated in a range of
coccolithophore species and strains (Buitenhuis et al., 2008), and seems to be a critical,
if not the most important, control on the biogeographical distribution of coccolithophore
species in the Southern Ocean (Trull et al., 2018). This pronounced latitudinal change in
coccolithophore assemblage composition contrasts with the little longitudinal variability
between the subantarctic SOTS and SAM sites (Fig. 3). These observations underscore
the role of circumpolar fronts as natural physical barriers for plankton species distribution
in the Southern Ocean (Medlin et al., 1994; Boyd, 2002; Cook et al., 2013).

Notably, the rare occurrence of the cold-water species *Coccolithus pelagicus* at

the SOTS and SAM sites contrasts with the high contribution of *C. pelagicus* to the living
coccolithophore communities in the subpolar and polar waters of the North Atlantic and
North Pacific oceans, where it is often the second most abundant species after *E. huxleyi*
(McIntyre and Bé, 1967; Baumann et al., 2000; Broerse et al., 2000a; Broerse et al.,
2000b; Ziveri et al., 2000). This important difference in species composition between
Northern and Southern hemisphere subpolar ecosystems could have important
implications in the calibration of the satellite PIC signal in the Southern Ocean. A recent
study by Trull et al. (2018) comparing satellite and shipboard observations identified a
substantial over-estimation of coccolithophore PIC in the Southern Ocean waters by the
NASA satellite ocean-colour-based PIC algorithm. Since satellite reflectance
observations are mainly calibrated against Northern Hemisphere PIC results (Balch et al.,
2011; Balch et al., 2016), we speculate that differences in the coccolithophore assemblage
composition, and particularly, differences in *C. pelagicus* numbers, could contribute to
the over-estimation of PIC concentrations by the satellite PIC algorithm in the Southern
Ocean. Indeed, the scaling of reflectance (in satellite images) to PIC (in ocean) is very
dependent on coccolith area:mass ratios (Gordon and Du, 2001; Balch et al., 2005).
*Coccolithus pelagicus* has remarkably heavier and thicker coccoliths (100-400 pg per
coccolith; Table 1) than *E. huxleyi* (~3 pg per coccolith), i.e. about 100 times heavier.
However, the average coccolith area of *C. pelagicus* is only about ten times greater than



that of *E. huxleyi*. Thus, this lack of proportional relationship between area and mass

between these species should be taken into consideration when calibrating the satellite

signals of coccolithophore-related PIC in the Southern Ocean.

**4.3 Coccolith calcite content of subantarctic coccolithophore species**

Multiple methodological biases associated with each of the methods used for

estimating coccolith calcite content (i.e. birefringence, morphometrics) could be invoked

to explain the different estimates observed for some of the species (see Young and Ziveri,

2000; Fuertes et al., 2014 and references therein). However, the fact that these

discrepancies vary greatly across species suggests that the composition of the crystal-

units of the coccoliths could be the most important factor causing these differences. All

the heterococcoliths of the species analysed are mainly composed of either V- or R-

calcite crystal units or a combination of both (Young et al., 2003; Table 1). R units are

characterized by sub-radial c-axes that are reasonably well measured by the birefringence

technique, but, the almost vertical optical axes of the V units (Young, 1992; Young et al.,

1999) make the same thickness less birefringent (Fuertes et al., 2014). Thus, it is likely

that differences in the birefringence properties of the R and V units could be responsible

for the different estimates provided by the two approaches. This is supported by our

results which show coccolith mass estimates of those species composed of R units, such

as *G. oceanica* and *Gephyrocapsa* spp. exhibit almost identical values with both

techniques (Table 1). In contrast, those species with coccoliths composed by a

combination of R and V units, such as *C. pelagicus*, *C. leptoporus*, *H. carteri* and *U.

sibogae,* display divergent mass estimates between approaches. The case of *E. huxleyi* is

more complex due to the large intraspecific genetic variability that results in substantial

differences in the profile and degree of calcification between specimens (Young and

Ziveri, 2000). Our birefringence mass estimate for *E. huxleyi* (2.67 ± 1.49 pg) is less than

one picogram lower than the mean range value calculated with the morphometric

technique (i.e. 1.81 ± 1.10 pg with an average $k_s$ value of all the morphotypes found at

the SAZ, i.e. $k_s = 0.0275$), but identical to the maximum (2.64 ± 1.60 pg; using $k_s = 0.04$).

These results suggest a reasonably good consistency between techniques for *E. huxleyi*.

Taking into consideration all the above, it is likely that the coccolith mass of some

species is underestimated by the birefringence technique, and therefore, the fractional

contribution of coccolithophores to total PIC using this approach should be taken as a

conservative estimate. Since both methods for estimating calcite content have inherent

uncertainties, the range of values provided by both techniques is used here as an
approximation of the fractional contribution of coccolithophores to total annual CaCO$_3$
export to the deep sea in the Australian and New Zealand sectors of the SAZ.

**4.4 Contribution of coccolithophores to carbonate export in the Australian-New**
**Zealand sectors of the Southern Ocean**
The magnitude of the total PIC export in the subantarctic waters was similar
between the SOTS and SAM sites (range 14-17 g m$^{-2}$ yr$^{-1}$), and thus slightly above the
global average (11 g m-2 yr-1; Honjo et al., 2008). Our estimates indicate that
coccolithophores are major contributors to CaCO$_3$ export in the Australian and New
Zealand waters of the SAZ, accounting for 40-60% and 15-25% of the annual CaCO$_3$
export, respectively (Fig. 5). Heterotrophic calcifiers, mainly planktonic foraminifera
(Salter et al., 2014), must therefore account for the remainder of the annual CaCO$_3$ export
at both sites. Previous work on foraminifera fluxes in our study regions allows an
approximate estimate of the contribution of foraminifera to total CaCO$_3$ flux that can be
used to assess the validity of our estimates. Combining counts of foraminifera
shells (King and Howard, 2003) with estimates of their average shell weights (20-40 µg
per shell depending on size; Moy et al., 2009) suggests contributions of 1/3 to 2/3 of
planktonic foraminifera to the total CaCO$_3$ flux in the Australian SAZ (Trull et al., 2018).
In the subantarctic waters south of New Zealand, Northcote and Neil (2005) estimated
that planktonic foraminifera accounted for about 78-97% of the total CaCO$_3$. Thus
estimations of the contribution of heterotrophic calcifiers to total carbonate in both study
regions are in reasonable agreement with our coccolithophore CaCO$_3$ estimates at both
sites. The lower contribution of coccolithophores to CaCO$_3$ export at the SAM site in
comparison with that of SOTS may be explained by differences in the ecosystem structure
between sites. Algal biomass accumulation in the surface waters of the SAM region
(average chlorophyll-*a* concentration between 2002 and 2018 is 0.31 mg m$^{-3}$) is
substantially higher than that at SOTS (0.23 mg m$^{-3}$). We speculate that the higher
abundance of non-calcareous phytoplankton (e.g. diatoms) in the subantarctic waters
south of New Zealand could simultaneously reduce coccolithophore biomass through
resource competition (Quéré et al., 2005; Sinha et al., 2010) while stimulating
foraminifera growth (Schiebel et al., 2017). The combination of both factors could be
responsible for the lower coccolithophore productivity at the SAM site despite similar
total CaCO$_3$ export. Assuming that both the SOTS and SAM sites can be considered



representative of a broad longitudinal swath of the SAZ south of Australia and New
Zealand (ca. 1% of areal extent of the global ocean), the coccolithophore $CaCO_3$ export
in these two regions together account for approximately 0.4 T $C_{inorg}$mol yr$^{-1}$. This value
represents approximately 1% of the global annual PIC export to the deep ocean (Honjo et
al., 2008) and underscores the notion that the high nutrient low-chlorophyll waters of the
circumpolar SAZ should not be taken as indicative of low biological activity or export.
Our results indicate that although *E. huxleyi* overwhelmingly dominates the
coccolithophore sinking assemblages at both study sites, other species with lower relative
contribution but substantially heavier coccoliths are more important contributors to the
annual coccolithophore $CaCO_3$ export budget (Fig. 3). Particularly relevant is the case of
*C. leptoporus* that despite its relatively low abundance (~ 10% of the annual assemblage
at both sites; Fig. 3), it accounts for between 30-50% and 60-70% of the annual
coccolithophore-$CaCO_3$ export at the SOTS and SAM sites, respectively (Fig. 3).
Similarly, other species with heavy coccoliths, such as *H. carteri* and *C. pelagicus*, are
important contributors to the annual coccolithophore PIC export to the deep sea (up to
~30% and ~10% of the annual coccolithophore PIC, respectively) despite their low annual
relative abundance (<2% at both sites) (Fig. 3). These results serve as an important
reminder that it is often not the most abundant species, but rather the largest
coccolithophore species that account for the greatest contribution to coccolithophore
$CaCO_3$ production and export (Young and Ziveri, 2000; Baumann et al., 2004; Daniels et
al., 2016).
The important contribution made by the coccolithophore community in setting the
magnitude of carbonate production and export to the deep sea is evidenced when we
compare the coccolith and total $CaCO_3$ fluxes of the SOTS sediment trap with those
deployed in the AZ along the 140°E meridian (Fig. 1). Although both total and
coccolithophore $CaCO_3$ export decrease with increasing latitude these changes are largely
uneven. While total $CaCO_3$ decreases two-fold from the SAZ to the AZ, coccolithophore
$CaCO_3$ export decreases 28-fold (Supplement Figure 2). This lack of proportional
latitudinal change can be attributed to two main factors. First, subantarctic
coccolithophore populations are diverse and relatively rich in species with large and
heavy coccoliths such as *C. leptoporus* or *H. carteri* that account for a large fraction of
the annual carbonate production and export. South of the PF, assemblages become
monospecific, or nearly monospecific, dominated by the small and relatively lightly





calcified *E. huxleyi*. Second, latitudinal variations in the abundance of heterotrophic
calcifiers (mainly foraminifera but also pteropods) must play a major role in modulating
the observed variations in CaCO$_3$ export. In particular, our data suggests that the
fractional contribution of heterotrophic calcifiers to CaCO$_3$ production increases from
~40-60 % in the Australian SAZ to up to 95% in the AZ (Rigual Hernández et al., 2018).
This pattern is consistent with previous shipboard and sediment trap studies that reported
higher abundances of planktonic foraminifera at the PFZ and AZ compared to that of the
SAZ in the Australian sector (King and Howard, 2003; Trull et al., 2018). Controls on the
biogeographic distribution of foraminifera species are complex and beyond the scope of
this paper, however, we provide a few observations. Both temperature and diet are critical
factors controlling the spatial distribution of planktonic foraminifera species. In
particular, the lower temperatures south of the SAF seem to favour the development of
*Neogloboquadrina pachyderma* sin. and *Turborotalita quinqueloba* as indicated by the
high abundance of these species in the PFZ (> 80% of the annual integrated flux for both
species together; King and Howard, 2003). Additionally, the dramatically different algal
communities dwelling in each zonal system may also play a role in planktonic
foraminifera species distributions. In particular, diatoms can account for a major part of
the diet of some foraminifera species, including *N. pachyderma* (Schiebel and Hemleben,
2017). Therefore, it is likely that the preferential grazing on diatoms of some foraminifera
species may play an important role in the increase of foraminifera CaCO$_3$ production
moving poleward.

**623 4.5 Future predictions of coccolithophore community response to environmental**

**624 change in the subantarctic zone**

The response of *E. huxleyi* to environmental change has been extensively studied
in laboratory experiments (Meyer and Riebesell, 2015; Müller et al., 2015; Feng et al.,
2017) and the available information is sufficient to propose possible changes of its niche
and calcification in the Southern Ocean, as discussed in detail in Trull et al. (2018). Due
to the ubiquity and abundance of *E. huxleyi*, the ecophysiology of this species is often
regarded as typical of all coccolithophores. However, *E. huxleyi* is rather different from
most other coccolithophore species in that its physiological adaptations place it in the
upper limit of the r-K ecological gradient of these organisms (i.e. an opportunistic
species), while most of the other species are located at the opposite end of the spectrum



(i.e. conservative or K-selected species) (Probert and Houdan, 2004). Our results
demonstrate that *E. huxleyi* plays an important, but not dominant role in CaCO₃ export,
with other species such as *C. leptoporus, H. carteri* or *C. pelagicus* making a larger
contribution to the annual CaCO₃ export in the SAZ (Fig. 3). Therefore, it is of critical
importance to evaluate how these other biogeochemically important coccolithophore
species will respond to projected climate-induced changes in the Southern Ocean. Here,
we now assess the response of large coccolithophore species to projected changes in
temperature and carbonate chemistry, that have been highlighted among the most
important environmental stressors expected to impact Southern Ocean coccolithophore
physiological rates (Müller et al., 2015; Charalampopoulou et al., 2016; Feng et al., 2017;
Trull et al., 2018).

The Southern Ocean is warming rapidly (Gille, 2002; Böning et al., 2008), largely

due to the southward migration of the ACC fronts (Sokolov and Rintoul, 2009). Only
between 1992 and 2007 the position of Southern Ocean fronts shifted by approximately
60 km to the south (Sokolov and Rintoul, 2009) and this trend is expected to continue
throughout the next century (Rintoul et al., 2018). Therefore, it is likely that any further
southward migration of ACC fronts will be coupled with an expansion of subantarctic
coccolithophore species towards higher latitudes. The poleward expansion of *E. huxleyi*
geographic range has already been suggested in the Southern Ocean (Cubillos et al., 2007;
Winter et al., 2014; Charalampopoulou et al., 2016) and it also appears to be occurring in
the North Atlantic (Rivero-Calle et al., 2015). Given the important contribution of large
subantarctic coccolithophore species to CaCO₃ export, the expansion of their ecological
niche could result in a substantial increase in CaCO₃ production and export in the
Southern Ocean. However, this may not be the future scenario for the SAZ southeast on
New Zealand, where bathymetry strongly controls the location of ocean fronts (Fernandez
et al., 2014; Chiswell et al., 2015). If the fronts are bathymetrically 'locked', then the
SAZ will not expand in areal extent, although the region is still predicted to undergo
significant physical, biogeochemical and biological changes (Law et al., 2017) that will
have likely flow-on effects on coccolithophore productivity and export (Deppeler and
Davidson, 2017).

The available carbonate chemistry manipulation experiments with *C. leptoporus*

have come to different conclusions. While some studies identified an increase in coccolith
malformations with increasing CO₂ concentrations (Langer et al., 2006; Langer and Bode,
2011; Diner et al., 2015), another study (Fiorini et al., 2011) reported no changes in the



calcification of *C. leptoporus* at elevated $pCO_2$. Interestingly, *C. leptoporus* did not
experience changes in its photosynthesis rates over the tested $CO_2$ range in any of the
aforementioned studies. The most likely explanation for the different results between the
studies is a strain-specific variable responses to changing carbonate chemistry (Diner et
al., 2015). Strain-specific variability in response to changing carbonate chemistry has
been previously reported in other coccolithophores, such as *E. huxleyi* (Langer et al.,
2009; Müller et al., 2015), and therefore it is likely that this also occurs in other species.
Given the fact that Southern Ocean fronts act as barriers for species distributions and gene
flows (Medlin et al., 1994; Patarnello et al., 1996; Thornhill et al., 2008; Cook et al.,
2013), it is possible that the subantarctic *C. leptoporus* populations exhibit a different
ecophysiology than those used in the above mentioned laboratory experiments. Prediction
of the responses of *H. carteri* and *C. pelagicus* is even more challenging due to the lack
of experiments testing the response of these species to changing seawater carbonate
chemistry. The only available insights in the response of one of these species to ocean
acidification are found in the fossil record. Both Gibbs et al. (2013) and O'Dea et al.
(2014) reconstructed the evolution of *C. pelagicus* populations during the Palaeocene-
Eocene Thermal Maximum (PETM), a period arguably regarded as the best geological
approximation of the present rapid rise in atmospheric $CO_2$ levels and temperatures.
These studies concluded that *C. pelagicus* most likely reduced its growth rates and
calcification during this period. This limited number of studies suggest that the ongoing
ocean acidification in the Southern Ocean could potentially have a negative impact on the
physiological rates of *C. leptoporus* and *C. pelagicus* while the effect on *H. carteri* is
unknown. Physiological response experiments (e.g. Müller et al., 2015) with Southern
Ocean strains of *C. leptoporus*, *H. carteri* and *C. pelagicus* are, therefore, urgently needed
to be able to quantify the effect of projected changes in oceanic conditions in the SAZ on
their physiological rates and consequent effects on carbon cycling in the Southern Ocean.

Our synthesis suggests opposing influence of environmental stressors on

subantarctic coccolithophore populations. Poleward migration of fronts will likely
increase coccolithophore $CaCO_3$ production in the Southern Ocean, while changes in
carbonate chemistry speciation will reduce growth rates of subantarctic coccolithophores.
It seems possible that coccolithophores will initially expand southward as waters warm
and fronts migrate, but then eventually diminish as acidification overwhelms those
changes.






## Acknowledgments

This project has received funding from the European Union's Horizon 2020 research and innovation programme under the Marie Skłodowska-Curie grant agreement number 748690 – SONAR-CO2 (ARH, JAF and FA). The SOTS mooring work was supported by IMOS, the ACE CRC, and the Australian Marine National Facility. The work at SAM was supported by funding provided by the New Zealand Ministry of Business, Innovation and Employment and previous agencies, and most recently by NIWA's Strategic Science Investment Fund. NIWA is acknowledged for providing capital grants for mooring equipment purchases, and thanks to all the NIWA scientists, technicians and vessels staff, who participated in the New Zealand biophysical moorings programme (2000-12). Cathryn Wynn-Edwards (IMAS) provided support in sample splitting/processing and laboratory analysis. Satellite Chlorophyll-*a* and PIC data sets were produced with the Giovanni online data system, developed
and maintained by the NASA GES DISC.

## Author contributions

TWT, SDN, DMD and LN planned and performed the field experiment. ARH led the coccolithophore study and performed sample processing and microscopy and image analyses. AMB and ARH performed SEM analyses. ARH and SN preformed numerical analyses. ARH wrote the paper with feedback from all authors.



**Competing interests**

The authors declare no competing interests.

**Data Availability**

Morphometric data of major coccolithophore species generated during the current study are listed in Table 1, while species relative abundance and species fluxes (plotted in Supplement Figure 1) will be publicly available through the Australian Antarctic Data Centre (link to be included before publication).

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
