# Peer review of "Coccolithophore biodiversity controls carbonate export in the Southern Ocean"

_Biogeosciences, 2019_

## Referee Comment (RC1) · Griet NEUKERMANS (Referee) · 23 Oct 2019

This paper is a very useful and original contribution to our understanding of how coccolithophore diversity shapes carbonate export in the Southern Ocean based on time series of sediment trap data. The paper is a pleasure to read: very well written, well structured, comprehensive, clear, and concise, with high-quality figures, and in-depth discussion. I highly recommend publication of this work in Biogeosciences. Congratulations to the authors for this very nice piece of work. I only have a few very minor comments that may improve the paper.

P3L87-89: replace "satellite reflectance observations" with "ocean color satellite re-

flectance observations" to precise that it is the fraction of incoming VISIBLE and NEAR-INFRARED solar radiation that is reflected from the ocean surface. Add reference (Balch et al., 2005) (Gordon et al., 2001) at the end of the sentence. These are the NASA standard algorithms for PIC retrieval.

P4L111: reference for representativeness is missing

P5L132: remove "that".

P6 Figure 1: STZ not included in legend.

P8L234: Can you briefly explain the method to calculate daily fluxes?

P9L253-255. I strongly appreciate the authors obtained two independent estimates of coccolith fluxes based on the birefringence and morphometric methods, each with their own advantages and disadvantages.

P10L294: Can you briefly explain why you think that the finding of <5% error on DSL estimates from polarization would apply to other species than the one tested?

Materials and Methods section: I think you should add a section on the ocean colour satellite data treatment. Which data did you use? Figure 2 suggests you used weekly data for PIC but monthly for Chla? Why not the same temporal resolution? Did you use multisensor merged products (such as GlobColour?)? Did you do any spatial averaging and how did you compute the weekly averages?

P12 Figure legend: specify "ocean color satellite-derived". Panel b, please add Chla data for October/November to see the potential rise in Austral spring. Can you present PIC and Chla data at the same temporal resolution? That would make sense.

P12 Figure 2: panel c at 61S is missing.

P16L429: the secondary maximum of satellite PIC might be an artefact of satellite data treatment, but it's hard to say, since that critical information is missing from the manuscript Materials and Methods. . .

P14L377: Not clear what you mean with total CaCO3 export in Fig. 5. Is this the combined export of coccos and forams? If yes, how did you quantify foram export? I suggest you also explain total CaCO3 in the Figure legend.

P17L436: it may also be a foraminiferan signal, see for example (Rembauville et al., 2016).

P18L497 etc.: The satellite PIC algorithm has indeed been calibrated in Northern hemisphere waters, where E. huxleyi greatly outnumbered other coccolithophore species, which is also the case in your study areas. In fact, the satellite signal (which is proportional to the particulate backscattering coefficient) is more sensitive to the concentration of E. huxelyi-sized particles, compared to larger, less abundant cocco species. Indeed, if larger, much heavier species are more prevalent in the Northern hemisphere waters, where the conversion factor for backscatter to PIC is calibrated, then this would lead to an overestimation of PIC in any waters where larger species are less prevalent. Put in other words, the conversion factor of backscatter to PIC is dependant on the size of the calcite particles. An alternative explanation for the overestimation of PIC is Southern Ocean waters is the contribution of bubbles to the backscattering coefficient.

P23L654: poleward expansion of E. huxleyi to the Arctic has also been demonstrated by (Neukermans et al., 2018)

P24L664 etc.: see also recent review in (Krumhardt et al., 2017)

References:

Balch, W. M., Gordon, H. R., Bowler, B. C., Drapeau, D. T., and Booth, E. S. (2005). Calcium carbonate measurements in the surface global ocean based on Moderate-Resolution Imaging Spectroradiometer data. J. Geophys. Res. 110, C07001. doi:10.1029/2004JC002560.

Gordon, H. R., Boynton, G. C., Balch, W. M., Groom, S. B., Harbour, D. S., and Smyth, T. J. (2001). Retrieval of coccolithophore calcite concentration from SeaWiFS Imagery.

Geophys. Res. Lett. 28, 1587–1590. doi:10.1029/2000GL012025.

Krumhardt, K. M., Lovenduski, N. S., Iglesias-Rodriguez, M. D., and Kleypas, J. A. (2017). Coccolithophore growth and calcification in a changing ocean. Prog. Oceanogr. 159, 276–295. doi:10.1016/j.pocean.2017.10.007.

Neukermans, G., Oziel, L., and Babin, M. (2018). Increased intrusion of warming Atlantic water leads to rapid expansion of temperate phytoplankton in the Arctic. Glob. Chang. Biol. 24, 2545–2553. doi:10.1111/gcb.14075.

Rembauville, M., Meilland, J., Ziveri, P., Schiebel, R., Blain, S., and Salter, I. (2016). Planktic foraminifer and coccolith contribution to carbonate export fluxes over the central Kerguelen Plateau. Deep Sea Res. Part I Oceanogr. Res. Pap. 111, 91–101. doi:10.1016/J.DSR.2016.02.017.

---

## Referee Comment (RC2) · Alex Poulton (Referee) · 1 Nov 2019

The manuscript by Rigual Hernandez et al. represents a comprehensive study of species-specific fluxes of coccolithophore-derived CaCO3 fluxes to the deep-sea in the Southern Ocean. The manuscript is well written and easy to follow, and provides several new insights into the important role of numerically rare coccolithophore species with high relative coccolith and cellular CaCO3 content. Such understanding has been well recorded in terms of production and export in northern polar and sub-polar waters, but the manuscript by these authors reveals the importance of this processes in the Australian-New Zealand sector of the Southern Ocean. There are no significant is-

sues with the methods or conclusions, and only a few points that need clarity or further referencing.

Ln 39: 'E. huxleyi dominates remote sensing images as a result of higher cell abundance and detachment of its small coccoliths.' This is an oversimplification and ignores the vital role of the characteristic light-scattering properties and size of E. huxleyi coccoliths, in addition to its tendency to shed coccoliths and characteristic bloom formation.

Ln 56-57: 'decline in saturation state of carbonate minerals in seawater makes the biological precipitation of carbonate difficult and increases dissolution rates of their shells or skeletons'. Current theoretical consensus of the response of coccolithophores to carbonate chemistry (e.g. Bach et al., 2015) specifically relates their internal calcification to substrate availability (HCO3-) and inhibition by proton (H+) concentrations; i.e. different carbonate chemistry parameters than inferred in the text (i.e. CaCO3 saturation state).

Bach et al. (2015). A unifying concept of coccolithophore sensitivity to changing carbonate chemistry embedded in an ecological framework. Progress in Oceanography, 135, 125-138.

Ln 92-95: As well as recent work by Trull et al. (2018) showing that satellite ocean-colour based PIC estimates can be unreliable in Antarctic waters, should also cite Holligan et al. (2010) which came to the same conclusion earlier.

Ln 131-132: 'which that', delete one or the other, both not necessary.

Ln 294-295: 'For the ks value of each taxa, data from the literature were (Table 1)' – sentence not finished.

Ln 329: Missing word – 'later' at end of sentence 'i.e. approximately eight months <later> (Fig. 2).'

Fig. 2. Would it not be better to make the y-axis on these plots the same scale?

Ln 417-419: This is an interesting point, as it is similar to loss terms found specifically for coccolithophores from microzooplankton grazing in the temperate N Atlantic setting (60-80%; Mayers et al., 2019).

Mayers et al. (2019). Growth and mortality of coccolithophores during spring in a temperate Shelf Sea (Celtic Sea, April 2015). Progress in Oceanography 177, 1010928.

Ln 490-492: Again, although Trull et al. (2018) recently identified over-estimate of coccolithophore PIC in the Southern Ocean by the NASA satellite ocean-colour-based PIC algorithm, this was examined earlier by Holligan et al. (2010). In the case of Holligan et al. (2010), the difference was attributed to the lower coccolith and cell $CaCO_3$ content of E huxleyi found in the S Atlantic (Scotia Sea). This is in general agreement with the reasoning suggested here (i.e. issues over the coccolith specific-area:mass ratios for the dominant reflective particles), though differs over whether this is considered a problem with E huxleyi or C pelagicus (or other species with high coccolith $CaCO_3$ content).

Ln 570: Should the units not be 0.4 Tmol C yr-1?

---

## Author Comment (AC1) · 22 Nov 2019

We sincerely thank Dr Griet Neukermans (reviewer 1) for her positive and constructive comments on our manuscript that have helped to improve the paper. We have carefully considered all her comments and addressed each of them as outlined below.

R1-Cx : Referee comment, R1-Rx: authors response.

R1-C1: This paper is a very useful and original contribution to our understanding of how coccolithophore diversity shapes carbonate export in the Southern Ocean based on time series of sediment trap data. The paper is a pleasure to read: very well written, well structured, comprehensive, clear, and concise, with high-quality figures, and in-depth discussion. I highly recommend publication of this work in Biogeosciences. Congratulations to the authors for this very nice piece of work. I only have a few very minor comments that may improve the paper.

R1-R1: We sincerely thank reviewer #1 for the careful reading of our manuscript and constructive criticisms and comments that helped to improve the manuscript. We have carefully considered all her comments and have addressed each of her concerns as outlined below.

R1-C2: P3L87-89: replace "satellite reflectance observations" with "ocean color satellite reflectance observations" to precise that it is the fraction of incoming VISIBLE and NEARINFRARED solar radiation that is reflected from the ocean surface. Add reference (Balch et al., 2005) (Gordon et al., 2001) at the end of the sentence. These are the NASA standard algorithms for PIC retrieval.

R1-R2: Corrected according to reviewer 1's suggestion.

R1-C3: P4L111: reference for representativeness is missing

R1-R3: Please note that a detailed explanation of the representativeness of the SOTS and SAM sites was explained later in the text (section 2.2). In the new version of the manuscript we refer to section 2.2 in the line indicated by the reviewer.

R1-C4: P5L132: remove "that".

R1-R4: Corrected according to reviewer 1's suggestion.

R1-C5: P6 Figure 1: STZ not included in legend.

R1-R5: Subtropical Zone - STZ has been included in the legend following reviewer 1's suggestion.

R1-C6: P8L234: Can you briefly explain the method to calculate daily fluxes?

R1-R6: The method employed to estimate coccolith and coccosphere fluxes has been included in the new version of the manuscript (lines 256-263 of the corrected version of the manuscript).

R1-C7:P9L253-255. I strongly appreciate the authors obtained two independent estimates of coccolith fluxes based on the birefringence and morphometric methods, each with their own advantages and disadvantages.

R1-R7: We appreciate reviewer 1's supportive comment. Since both techniques have associated errors, we decided to present both estimates (in the manuscript). Interestingly, in spite of some variability between techniques the general conclusions would remain similar to using any of the techniques individually.

R1-C8:P10L294: Can you briefly explain why you think that the finding of <5% error on DSL estimates from polarization would apply to other species than the one tested?

R1-R8: E. huxleyi overwhelmingly dominated the coccolithophore assemblages in all the samples analysed. Given the very low number of coccoliths of the rest of the coccolithophore species, it was almost impossible to find a representative number of individuals of for most of the "secondary" species in the same sample in order to statistically compare both microscopy techniques. Please note that even when a coccolith of a given species is found under the SEM, it can not always be measured because its position is not always adequate (e.g. they are often tilted or partially covered by other phytoplankton or detritus). Based on this, we decided to measure C. leptoporus because it was the second most abundant species, and therefore, statistical comparison between populations measured under the Light Microscope (LM) and SEM was possible. The subtle differences between coccolith distal length measurements are most likely due to the fact that the peripheral limit of the coccolith shield is not as sharp under the LM as is the case for SEM images. It follows that differences in coccolith measurements between SEM and LM techniques will be probably similar or smaller in the case of larger species. Since the majority of coccolith species identified in the

current study display a similar (e.g. Gephyrocapsa oceanica, Syracosphaera pulchra, Umbellosphaera tenuis and Umbilicosphaera sibogae) or larger size (e.g. Coccolithus pelagicus and Helicosphaera carteri) than C. leptoporus, it can be assumed that the <5% error on DSL estimates for C. leptoporus is applicable to the rest of the species found in the current study.

R1-C9: Materials and Methods section: I think you should add a section on the ocean colour satellite data treatment. Which data did you use? Figure 2 suggests you used weekly data for PIC but monthly for Chla? Why not the same temporal resolution? Did you use multisensor merged products (such as GlobColour?)? Did you do any spatial averaging and how did you compute the weekly averages?

R1-R9: Corrected according to reviewer 1's suggestion. A new subsection called "2.8 Remotely sensed chlorophyll-a and PIC concentrations" has been included in the new version of the manuscript describing how we obtained and processed the Chl-a and PIC satellite data used in the manuscript. Weekly Chl-a data is now plotted in the graphs. Additionally, in order to support our statements in section 4.1 of the discussion, CaCO3 fluxes registered by the traps have also been included in Figure 2.

R1-C10: P12 Figure legend: specify "ocean color satellite-derived". Panel b, please add Chla data for October/November to see the potential rise in Austral spring. Can you present PIC and Chla data at the same temporal resolution? That would make sense. R1-R10: Corrected according to reviewer 1's suggestion. As mentioned in the previous comment weekly Chl-a data is now plotted in Figure 2. Moreover, data for the month of November is now included in figure 2.

R1-C11: P12 Figure 2: panel c at 61S is missing.

R1-R11: The figure caption erroneously mentioned the 61°S site (the figure caption corresponds to an earlier version of the manuscript where data from the 61°S was presented in the graph). In the new version of the manuscript this information is not required. Therefore, the reference to the 61°S site in the caption of Figure 2 has been

deleted.

R1-C12:P16L429: the secondary maximum of satellite PIC might be an artefact of satellite data treatment, but it's hard to say, since that critical information is missing from the manuscript Materials and Methods...

R1-R12: As mentioned above (see R1-R9), a new subsection called "2.8 Remotely sensed chlorophyll-a and PIC concentrations" has been included in the manuscript. It is important to note that the PIC satellite signal for the grid area considered representative of the SAM station (coordinates 47-45° S and 171°E-179°W) was almost identical to that of a smaller area over the SAM site (47-45° S, 177.5-179.5°E). An alternative explanation of the secondary PIC maximum (i.e. possibility of storm-induced bubbles) has been included in the text. See section 4.1 of the new version of the manuscript.

R1-C13: P14L377: Not clear what you mean with total CaCO3 export in Fig. 5. Is this the combined export of coccos and forams? If yes, how did you quantify foram export? I suggest you also explain total CaCO3 in the Figure legend. R1-R13: Both figure and figure caption have been modified in order to make clear that annual total CaCO3 export (represented by yellow bars in Figure 5) refers to the total amount of CaCO3 collected by the traps determined chemically (as explained in section 2.4) while the clear and dark blue bars represent the two different estimates of the contribution of CaCO3 based on birefringence and morphometric techniques, respectively.

R1-C14: P17L436: it may also be a foraminiferan signal, see for example (Rembauville et al., 2016).

R1-R14: We appreciate reviewer 1's suggestion. Indeed, we did consider the possibility that heterotrophic calcifying plankton such as planktonic foraminifera or pteropods could account for the secondary maximum in February-early-March. However, total CaCO3 fluxes recorded in the trap do not reflect an increase during this interval. Therefore, we believe this possibility is unlikely. Please note that in the new version of the manuscript total CaCO3 fluxes have been included in figure 2 (see also R1-R9).

R1-C15: P18L497 etc.: The satellite PIC algorithm has indeed been calibrated in Northern hemisphere waters, where E. huxleyi greatly outnumbered other coccolithophore species, which is also the case in your study areas. In fact, the satellite signal (which is proportional to the particulate backscattering coefficient) is more sensitive to the concentration of E. huxelyi-sized particles, compared to larger, less abundant cocco species. Indeed, if larger, much heavier species are more prevalent in the Northern hemisphere waters, where the conversion factor for backscatter to PIC is calibrated, then this would lead to an overestimation of PIC in any waters where larger species are less prevalent. Put in other words, the conversion factor of backscatter to PIC is dependant on the size of the calcite particles. An alternative explanation for the overestimation of PIC is Southern Ocean waters is the contribution of bubbles to the backscattering coefficient.

R1-R15: Our intention was to highlight that the different composition of coccolithophore assemblages between the Northern Hemisphere and Southern Ocean may contribute (only one factor among probably many) to the overestimation of PIC concentration in the Southern Ocean. In the new version of the manuscript this has been clarified and the possible influence of microbubbles to the backscattering coefficient has also been included.

R1-C16: P23L654: poleward expansion of E. huxleyi to the Arctic has also been demonstrated by (Neukermans et al., 2018)

R1-R16: We appreciate the new reference provided by reviewer 1. This study is now mentioned in the new version of the manuscript.

R1-C17: P24L664 etc.: see also recent review in (Krumhardt et al., 2017)

R1-R17: The reference mentioned by reviewer 1 has been included in the discussion (section 4.5).

---

## Author Comment (AC2) · 22 Nov 2019

We sincerely thank Dr Alex Poulton (reviewer 2) for the valuable comments and suggestions that have helped to improve the original version of the manuscript. We have carefully considered all his comments and have addressed each of his concerns as outlined below.

R2-Cx : Referee comment, R2-Rx: authors response.

R2-C1: The manuscript by Rigual Hernandez et al. represents a comprehensive study of species-specific fluxes of coccolithophore-derived CaCO3 fluxes to the deep-sea in

the Southern Ocean. The manuscript is well written and easy to follow, and provides several new insights into the important role of numerically rare coccolithophore species with high relative coccolith and cellular CaCO3 content. Such understanding has been well recorded in terms of production and export in northern polar and sub-polar waters, but the manuscript by these authors reveals the importance of this processes in the Australian-New Zealand sector of the Southern Ocean. There are no significant issues with the methods or conclusions, and only a few points that need clarity or further referencing.

R2-R1: We sincerely appreciate reviewer 2 for taking the time to carefully read the manuscript and providing valuable comments and references that helped to improve the manuscript.

R2-C2: Ln 39: 'E. huxleyi dominates remote sensing images as a result of higher cell abundance and detachment of its small coccoliths.' This is an oversimplification and ignores the vital role of the characteristic light-scattering properties and size of E. huxleyi coccoliths, in addition to its tendency to shed coccoliths and characteristic bloom formation.

R2-R2: The sentence referred by reviewer 2 has been replaced by the following: "This observation contrasts with the generally accepted notion that high PIC accumulations during the austral summer in the subantarctic Southern Ocean are mainly caused by E. huxleyi blooms.".

R2-C3: Ln 56-57: 'decline in saturation state of carbonate minerals in seawater makes the biological precipitation of carbonate difficult and increases dissolution rates of their shells or skeletons'. Current theoretical consensus of the response of coccolithophores to carbonate chemistry (e.g. Bach et al., 2015) specifically relates their internal calcification to substrate availability (HCO3-) and inhibition by proton (H+) concentrations; i.e. different carbonate chemistry parameters than inferred in the text (i.e. CaCO3 saturation state). Bach et al. (2015). A unifying concept of coccolithophore sensitivity

to changing carbonate chemistry embedded in an ecological framework. Progress in Oceanography, 135, 125-138.

R2-R3: The sentence highlighted by Reviewer 2 has been modified taking into consideration his suggestion and the reference of Bach et al. (2015) is now mentioned in the text (see lines first part of the introduction of the corrected version of the manuscript). Please note that in this part of the introduction we are talking in general about marine calcifying organisms, i.e. not specifically about coccolithophores.

R2-C4: Ln 92-95: As well as recent work by Trull et al. (2018) showing that satellite oceancolour based PIC estimates can be unreliable in Antarctic waters, should also cite Holligan et al. (2010) which came to the same conclusion earlier.

R2-R4: Corrected according to reviewer 2's suggestion. Holligan et al. (2010) paper is now mentioned together with Trull et al. (2018) in the new version of the manuscript.

R2-C5: Ln 131-132: 'which that', delete one or the other, both not necessary.

R2-R5: Corrected according to reviewer 1 and 2's suggestion.

R2-C6: Ln 294-295: 'For the ks value of each taxa, data from the literature were (Table 1)' – sentence not finished.

R2-R5: The sentence referred to by reviewer 2 has been modified. Now it reads: "For the ks value of each taxa, data from the literature was employed (Table 1)."

R2-C7: Ln 329: Missing word – 'later' at end of sentence 'i.e. approximately eight months <later> (Fig. 2).'

R2-R5: We intended to say that the period of elevated coccolith flux lasted about 8 months. However, this information is not of critical importance and therefore we have deleted the end of the sentence.

R2-C8: Fig. 2. Would it not be better to make the y-axis on these plots the same scale?

R2-R5: Corrected according to reviewer 2's suggestion. The y-axes have now the same scale in each station. Please note that in figure SAM site two axes (coccospheres and PIC) required different scale due to the different magnitude of these parameters compared to those of the SOTS site.

R2-C9: Ln 417-419: This is an interesting point, as it is similar to loss terms found specifically for coccolithophores from microzooplankton grazing in the temperate N Atlantic setting (60-80%; Mayers et al., 2019). Mayers et al. (2019). Growth and mortality of coccolithophores during spring in a temperate Shelf Sea (Celtic Sea, April 2015). Progress in Oceanography 177, 1010928.

R2-R9: We appreciate reviewer 2's suggestion. This is a good point that has been included in the new version of the manuscript (see section 4.1 of the new version of the manuscript).

R2-C10: Ln 490-492: Again, although Trull et al. (2018) recently identified overestimate of coccolithophore PIC in the Southern Ocean by the NASA satellite ocean-colour-based PIC algorithm, this was examined earlier by Holligan et al. (2010). In the case of Holligan et al. (2010), the difference was attributed to the lower coccolith and cell CaCO3 content of E huxleyi found in the S Atlantic (Scotia Sea). This is in general agreement with the reasoning suggested here (i.e. issues over the coccolith specific-area:mass ratios for the dominant reflective particles), though differs over whether this is considered a problem with E huxleyi or C pelagicus (or other species with high coccolith CaCO3 content).

R2-R10: We agree with reviewer 2. The text has been modified including Holligan et al. (2010) reference in the manuscript. Now it reads: "Since satellite reflectance observations are mainly calibrated against Northern Hemisphere PIC results (Balch et al., 2011; Balch et al., 2016), the lower the calcite content of dominant E. huxleyi morphotypes (B/C) in the Southern Ocean compared to their northern hemispheric counterparts has been suggested as a possible factor causing the over-estimation of
PIC concentrations in the Southern Ocean. Following this reasoning, we speculate that differences in other components of the coccolithophore assemblages, and particularly, differences in C. pelagicus numbers, could contribute to the over-estimation of PIC concentrations by the satellite PIC algorithm in the Southern Ocean. Indeed,..."

R2-C11: Ln 570: Should the units not be 0.4 Tmol C yr-1?

R2-R11: Reviewer 2 is correct, this error has been corrected in the new version of the manuscript.